# Development of Clinical Decision Models for the Prediction of Systemic Lupus Erythematosus and Sjogren’s Syndrome Overlap

**DOI:** 10.3390/jcm12020535

**Published:** 2023-01-09

**Authors:** Yan Han, Ziyi Jin, Ling Ma, Dandan Wang, Yun Zhu, Shanshan Chen, Bingzhu Hua, Hong Wang, Xuebing Feng

**Affiliations:** Department of Rheumatology and Immunology, The Affiliated Drum Tower Hospital of Nanjing University Medical School, Nanjing 210008, China

**Keywords:** systemic lupus erythematosus, Sjögren’s syndrome, overlap, prediction model

## Abstract

Objective: To explore the clinical features of patients with systemic lupus erythematosus and Sjögren’s syndrome overlap (SLE-SS) compared to concurrent SLE or primary SS (pSS) patients, we utilized a predictive machine learning-based tool to study SLE-SS. Methods: This study included SLE, pSS, and SLE-SS patients hospitalized at Nanjing Drum Hospital from December 2018 to December 2020. To compare SLE versus SLE-SS patients, the patients were randomly assigned to discovery cohorts or validation cohorts by a computer program at a ratio of 7:3. To compare SS versus SLE-SS patients, computer programs were used to randomly assign patients to the discovery cohort or the validation cohort at a ratio of 7:3. In the discovery cohort, the best predictive features were determined using a least absolute shrinkage and selection operator (LASSO) logistic regression model among the candidate clinical and laboratory parameters. Based on these factors, the SLE-SS prediction tools were constructed and visualized as a nomogram. The results were validated in a validation cohort, and AUC, calibration plots, and decision curve analysis were used to assess the discrimination, calibration, and clinical utility of the predictive models. Results: This study of SLE versus SLE-SS included 290 patients, divided into a discovery cohort (*n* = 203) and a validation cohort (*n* = 87). The five best characteristics were selected by LASSO logistic regression in the discovery cohort of SLE versus SLE-SS and were used to construct the predictive tool, including dry mouth, dry eye, anti-Ro52 positive, anti-SSB positive, and RF positive. This study of SS versus SLE-SS included 266 patients, divided into a discovery cohort (*n* = 187) and a validation cohort (*n* = 79). In the discovery cohort of SS versus SLE-SS, by using LASSO logistic regression, the eleven best features were selected to build the predictive tool, which included age at diagnosis (years), fever, dry mouth, photosensitivity, skin lesions, arthritis, proteinuria, hematuria, hypoalbuminemia, anti-dsDNA positive, and anti-Sm positive. The prediction model showed good discrimination, good calibration, and fair clinical usefulness in the discovery cohort. The results were validated in a validation cohort of patients. Conclusion: The models are simple and accessible predictors, with good discrimination and calibration, and can be used as a routine tool to screen for SLE-SS.

## 1. Introduction

Systemic lupus erythematosus (SLE) is a typical autoimmune connective tissue disease that can clinically accumulate in multiple organs throughout the body and is more prevalent in women of reproductive age [1,2]. Sjögren’s syndrome (SS) is a chronic inflammatory autoimmune disease mainly involving the exocrine glands. SS is clinically manifested as dryness of the mouth and eyes, but extra glandular organs may also be involved, either independently or in combination with other connective tissue diseases, including rheumatoid arthritis and systemic lupus erythematosus [3,4,5]. These two diseases often coexist in the same individual, which may affect the proper diagnosis and prognosis for these patients. The association of SLE and SS was first described in 1959, and SLE with SS is considered a distinct subtype of SLE [6]. In recent years, the number of patients with SLE and SS overlap has gradually increased, and studies have shown that 8.3–19.0% of SLE patients also have SS [7]. Another study in China even showed that 24 out of 55 SLE cases (43.6%) had secondary SS [8].

In addition to finding that the prognosis of patients with SLE-SS overlap is different from that of patients with a single disease, previous studies have also revealed that these patients have unique clinical characteristics [9,10,11]. Unfortunately, the conclusions in different reports are varied, and more than 10 types of clinical manifestations are involved, which is difficult to use for routine clinical implementation. Therefore, it is necessary to develop a practical and reliable screening technique to help identify patients with SLE-SS overlap early to improve their long-term outcome.

In this study, we retrospectively analyzed clinical data related to patients with SLE, SS, and SLE-SS. The risk factors affecting patients with SLE or SS in SLE-SS were extracted, and a decision model for predicting patients with SLE or SS versus SLE-SS was constructed. This model may provide a simple and effective tool for identifying patients with SLE-SS.

## 2. Materials and Methods

### 2.1. Patients and Controls

Data were retrospectively collected from 477 patients with SLE (*n* = 211), pSS (*n*= 187), and SLE-SS (*n* = 79) between December 2018 and December 2020 from the Department of Rheumatology and Immunology of Nanjing Drum Tower Hospital. The diagnosis of SLE was given according to the 1997 American College of Rheumatology (ACR) revised criteria for the classification of systemic lupus erythematosus [12], and the diagnosis of pSS was based on the 2002 American-European Consensus group criteria [13]. SLE-SS is defined as meeting the diagnostic criteria for both SLE and secondary SS, excluding patients with lymphoma, nodal disease, hepatitis virus infection, AIDS, radiation therapy, and anti-acetylcholine drugs. 

### 2.2. Data Collection

Data on demographic characteristics and clinical and laboratory findings were collected retrospectively. The clinical features assessed included fever, alopecia, asthenia, dry mouth, dry eye, photosensitivity, skin lesions, oral ulcer, Raynaud phenomenon, cavities, epistaxis, arthritis, interstitial lung disease, pulmonary arterial hypertension, and vasculitis. Laboratory findings included proteinuria, hematuria, anemia, leukocytopenia, thrombocytopenia, hypoalbuminemia, increased blood urea nitrogen, increased serum creatinine, increased erythrocyte sedimentation rate (ESR), increased C-reactive protein, hypocomplementemia, ANA, anti-dsDNA antibody, anti-SSA antibody, anti-Ro52 antibody, anti-SSB antibody, anti-Sm antibody, anti-RNP antibody, anti-cardiolipin antibody, rheumatoid factor (RF)-positive, and IgG elevation. Normal values for the laboratory tests were as follows: hemoglobin ≥ 110 g/L (female) or 120 g/L (male); platelets 100–300 × 10^9^/L; leukocytes 4–10 × 10^9^/L; serum albumin ≥ 35 g/L; blood urea nitrogen (BUN) ≤ 7.5 mmol/L; serum creatinine ≤ 133 μmol/L; erythrocyte sedimentation rate (ESR) ≤ 20 (female) or ≤15 mm/h (male); CRP ≤ 8 mg/L; complement C3 ≥ 0.8 g/L or C4 ≥ 0.2 g/L; ANA-negative (<1:40); anti-dsDNA-negative, anti-SSA-negative, anti-Ro52-negative, anti-SSB-negative, anti-Sm-negative, and anti-RNP-negative; anti-cardiolipin antibody < 12 U/mL or negative; RF < 20 IU/mL; IgG ≤ 16 g/L; and urine protein < 0.5 g/24 h or less than 2+. All the antibodies tested were IgG type, and positivity and negativity were defined according to the standard in the Drum Tower Hospital.

### 2.3. Statistical Analysis

Categorical variables are presented as numbers and standard frequencies and were tested by the χ^2^ test or Fisher’s exact test. Continuous variables are presented as medians and interquartile ranges (IQRs) and were compared using the Mann–Whitney U test. 

For SLE versus SLE-SS, a total of 290 patients were randomly assigned to the discovery cohort (*n* = 203) or the validation cohort (*n* = 87) using a computer program at a ratio of 7:3. Similarly, for SS versus SLE-SS, a total of 266 patients were randomly assigned to the discovery cohort (*n* = 187) or the validation cohort (*n* = 79) using a computer program at a ratio of 7:3. For the training group, the sample size followed the 10-fold criterion, that is, each prediction variable needs at least 10 observations to produce a reasonable and stable estimate [14,15]. In this study, 5 predictive variables were selected as the final model, requiring at least 50 observations and a sample size of at least 200 for the training group based on a 25% incidence. PASS 15 software (NCSS, LLC, Kaysville, Utah) was used to calculate the sample size of the verification group [16]. The minimum number of positive events was 10, the minimum number of negative events was 50, and the minimum sample size of the verification group was 60. A sufficient sample size is included in this study. We developed prediction tools using the discovery cohorts. A least absolute shrinkage and selection operator (LASSO) logistic regression model and 10-fold cross-validation were applied to select the best predictive features among the clinical characteristics and laboratory parameters in the discovery cohort that were statistically different between SLE and SLE-SS or SS and SLE-SS. The performance of the predictive tool was evaluated for discrimination, calibration, and clinical usefulness. To assess the discrimination of the predictive model, the discriminatory ability was measured using the area under the receiver operating characteristic (AUC) curve. Calibration curves were then plotted to evaluate the calibration effect of the SLE-SS prediction models. Decision curve analysis was used by quantifying the net benefits to determine the clinical utility of the SLE-SS predictive models.

All data analyses used R software (version 4.0.3). In R software, Packages (“rms”) and (“rmda”) were operated. All tests were two-sided, and a *p*-value < 0.05 was considered statistically significant.

## 3. Results

### 3.1. Patient Characteristics

For SLE versus SLE-SS, a total of 290 patients were included in this study, and a random sample at a ratio of 7:3 was divided into discovery (*n* = 203) and validation groups (*n* = 87). Regarding SS versus SLE-SS, a total of 266 patients were included in this study and were randomized in a 7:3 ratio to the discovery (*n* = 187) and validation groups (*n* = 79). A flow chart is shown in Figure 1. Appendix A summarizes the clinical characteristics and laboratory parameters of the discovery and validation groups. The characteristics of the two groups of patients were comparable. We also summarize the clinical characteristics and laboratory parameters of SLE, SS, and SLE-SS in the discovery and validation cohorts in Appendix A.

### 3.2. Development of the Prediction Tool for SLE or SLE-SS

The 11 candidate clinical and laboratory indicators that were statistically different between SLE-SS and SLE were reduced to the five most useful predictive markers by using LASSO logistic regression (Figure 2A,B). These characteristics included dry mouth, dry eye, anti-Ro52 positive, anti-SSB positive, and RF positive. The results of the logistic regression analysis for SLE versus SLE-SS are given in Table 1. We constructed prediction tools based on these factors for SLE in SLE-SS and visualized them as nomograms (Figure 3A).

Similarly, as shown in Figure 2C,D, LASSO logistic regression was used to reduce the 17 candidate laboratory indicators that were statistically different between SLE-SS and SS to the 11 most useful predictive markers including age at diagnosis (years), fever, dry mouth, photosensitivity, skin lesions, arthritis, proteinuria, hematuria, hypoalbuminemia, anti-dsDNA positive, and anti-Sm positive. The results of the logistic regression analysis of SS and SLE-SS are shown in Table 1. Based on these factors, we constructed a prediction tool for SS in SLE-SS and visualized it as a nomogram (Figure 3B).

### 3.3. Model Performance Assessment

The AUC of SLE versus SLE-SS and SS versus SLE-SS was 0.880 (95% CI: 0.826–0.934) and 0.970 (95% CI: 0.946–0.995), respectively, indicating that the predictive tool had good discrimination in the discovery cohort (Figure 4A,D). In this study, the calibration curves for both SLE versus SLE-SS and SS versus SLE-SS also showed good nomogram agreement in the discovery cohorts (Figure 4C,F). The DCA of the SLE versus SLE-SS prediction tool is presented in Figure 5A. The decision curve showed that if the threshold probability of a patient and a doctor was >5 and <78%, respectively, the use of this SLE versus SLE-SS nomogram to predict the risk of SLE-SS added more benefit than the scheme. The DCA of the SS versus SLE-SS prediction tool indicated that the prediction tool conferred more net benefits than the SLE-SS patients’ scheme across all threshold probabilities (Figure 5C). The validation cohort provided reconfirmation that the prediction tool had good discrimination (AUC: 0.865; 95% CI: 0.781–0.949, Figure 4B; AUC: 0.936; 95% CI: 0.888–0.984, Figure 4E), good calibration (Figure 4C,F), and clinical utility (Figure 5B,D).

## 4. Discussion

Previous studies have explored potential risk factors for SLE or SS in SLE-SS, including demographics, clinical characteristics, and laboratory parameters [10,11,17,18]. However, no studies have investigated models that incorporate multiple laboratory findings in clinical decision making. In this study, the SLE-SS overlapping disease prediction models were established using the minimum absolute contraction and selection operator (LASSO) logistic regression model between the candidate clinical features and laboratory parameters and were visualized as nomograms to help predict the risk of SLE or SS in SLE-SS.

The clinical features of SS versus SLE-SS, such as dry mouth, dry eye, and a unique autoantibody profile associated with pSS, were identified. In Yang’s study, we found that the frequency of dry mouth and dry eye was significantly higher in SLE-SS than in the SLE group [18]. Most previous studies have shown that anti-SSA/Ro and anti-SSB/La antibodies are significantly more positive in SLE-SS than in the SLE group [19]. Regarding RF positivity, Santos and Manoussakis et al. showed that RF positivity was significantly higher in both SLE-SS groups than in the SLE group [20]. Our results were largely consistent with previous studies. We conclude that dry mouth, dry eye, anti-Ro52 positive, anti-SSB positivity, and RF positivity are risk factors for SLE progressing to SLE-SS. Therefore, we should be aware of the development of combined SS when young SLE patients present with dry mouth, dry eye, or positive anti-Ro52, anti-SSB, or RF.

Meanwhile, patients with SLE-SS also have some clinical characteristics and autoantibody profiles specific to SLE. Szanto and Manoussakis et al. showed a higher rate of anti-dsDNA antibody positivity in SLE-SS than in the SS group and a significantly higher frequency of hypoalbuminemia in SLE-SS than in the SS group [10,11]. Patients with SLE-SS are also more likely to have renal involvement, such as proteinuria and hematuria. The study by Szanto and Yang et al. suggests that SLE-SS patients are more likely to develop renal damage than patients with SS [20,21]. In our study, we constructed a predictive model with risk factors such as proteinuria, hematuria, hypoalbuminemia, and positive anti-dsDNA antibodies, suggesting that when SS patients present with these symptoms, they are more likely to have combined SLE.

There are several limitations of this study. First, our collected data are retrospective, and various biases are inevitable due to the nature of the study itself, even though all three groups of patients were subjected to the same conditions and criteria to avoid selection bias and confounding factors. Second, the sample size was relatively small, and we welcome external validation in a wider population.

## 5. Conclusions

Based on the patient’s clinical symptoms, the development of SS should be monitored when SLE patients are positive for anti-RO52, anti-SSB, and RF. However, the presence of SS patients with albuminuria, hematuria, hypoproteinemia, positive anti-dsDNA, and positive anti-SM may indicate the presence of SLE. On the basis of clinical symptoms and serum indicators, this study established a risk prediction model for SLE or SS transitioning into SLE-SS with high accuracy, which may help with early diagnosis and the selection of appropriate treatment.

## Figures and Tables

**Figure 1 jcm-12-00535-f001:**
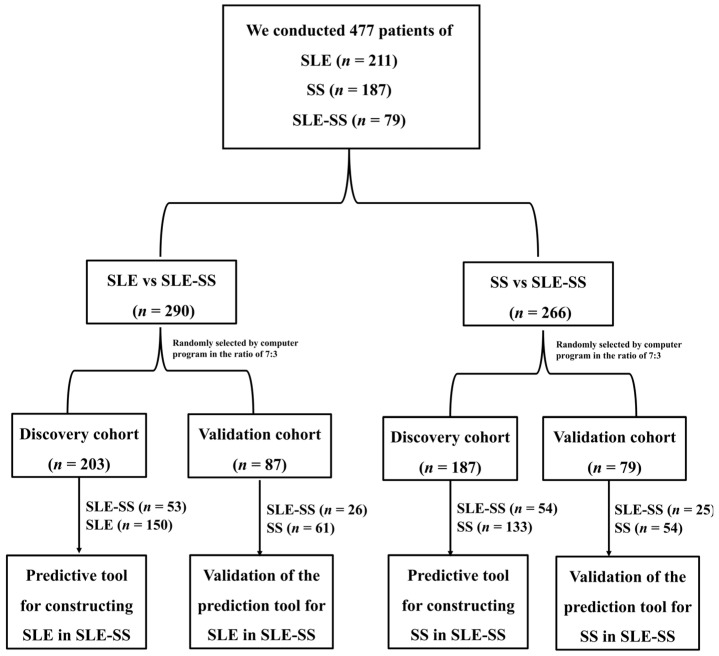
Flowchart for constructing the SLE or SS in the SLE-SS prediction tool.

**Figure 2 jcm-12-00535-f002:**
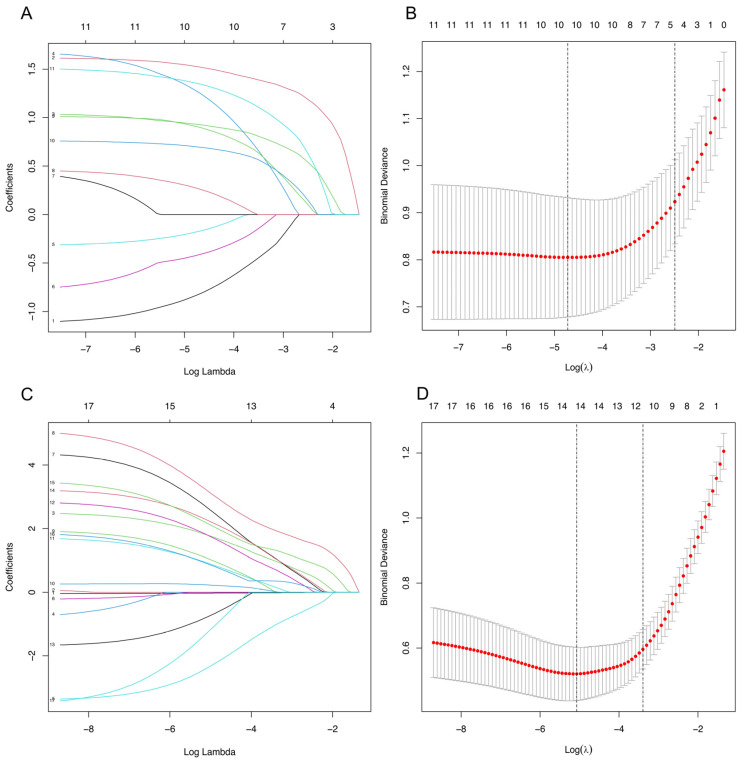
Clinical and laboratory parameter selection using the LASSO binary logistic regression model. (**A**,**B**) Results for the 11 variables included in the LASSO regression associated with SLE in SLE-SS and the corresponding coefficients for the corresponding variables at different values. (**C**,**D**) Results for the 17 variables included in the LASSO regression associated with SS and their corresponding coefficients for the corresponding variables at different values of coefficients. LASSO, minimum absolute shrinkage, and selection operators.

**Figure 3 jcm-12-00535-f003:**
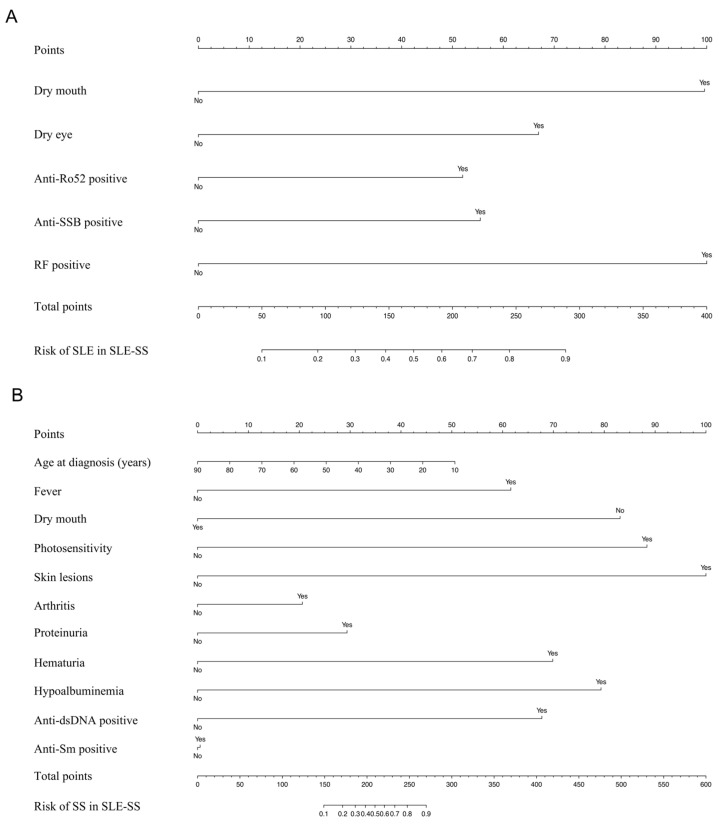
Nomogram prediction of SLE-SS. (**A**) The prediction model of SLE in SLE-SS was visualized as a nomogram. (**B**) The prediction model of SS in SLE-SS was visualized as a nomogram.

**Figure 4 jcm-12-00535-f004:**
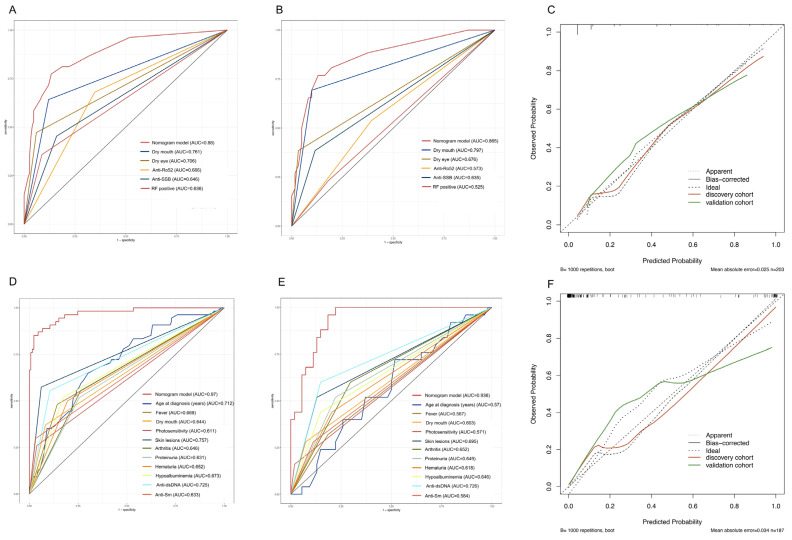
SLE-SS of predictive tool evaluation in SLE or SS patients. (**A**,**B**) ROC curves of SLE in SLE-SS based prediction tools in the discovery and validation cohorts, respectively. (**D**,**E**) ROC curves of SS in SLE-SS-based prediction tools in the discovery and validation cohorts, respectively. (**C**,**F**) Calibration plots for predicting SLE or SS in SLE-SS patients.

**Figure 5 jcm-12-00535-f005:**
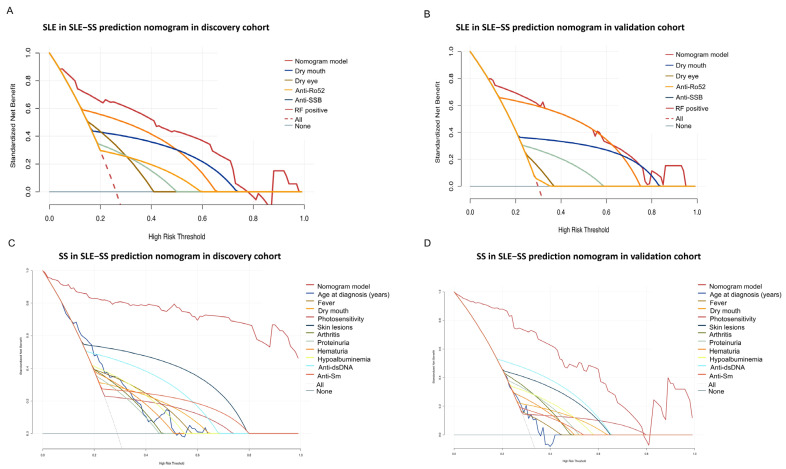
The SLE in the SLE-SS decision curves analysis in the discovery cohort (**A**) and validation cohort (**B**) assessed the clinical usefulness of the prediction model and other indicators. The SS in the SLE-SS decision curves analysis in the discovery cohort (**C**) and validation cohort (**D**) assessed the clinical usefulness of the prediction model and other indicators.

**Table 1 jcm-12-00535-t001:** Risk factors for patients with SLE or SS to develop SLE-SS.

	Intercept and Variable	Prediction Model		
		β	Odds Ratio (95% CI)	*p*-Value
SLE develop SLE-SS	Dry mouth	1.83	6.24(2.00–19.80)	0.00
	Dry eye	1.23	3.42(0.95–12.89)	0.06
	Anti-Ro52 positive	0.96	2.60(1.13–6.18)	0.03
	Anti-SSB positive	1.02	2.77(1.12–6.91)	0.03
	RF positive	1.84	6.29(2.37–17.32)	0.00
	Age at diagnosis (years)	−0.02	0.98(0.94–1.02)	0.28
SS develop SLE-SS	Fever	2.24	9.40(2.17–48.81)	0.00
	Dry mouth	−3.02	0.049(0.01–0.31)	0.00
	Photosensitivity	3.22	24.94(2.72–363.01)	0.01
	Skin lesions	3.64	38.04(6.25–341.38)	0.00
	Arthritis	0.75	2.12(0.53–8.82)	0.29
	Proteinuria	1.07	2.91(0.68–13.06)	0.15
	Hematuria	2.54	12.0(2.62–78.66)	0.00
	Hypoalbuminemia	2.89	17.95(4.07–106.35)	0.00
	Anti-dsDNA positive	2.46	11.75(2.82–61.23)	0.00
	Anti-Sm positive	0.02	0.98(0.12–8.81)	0.99

β is the regression coefficient. OR: Odds ratio; CI: Confidence interval.

## Data Availability

Data are available on request from the corresponding author.

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
