# Peer review of "Development of Clinical Decision Models for the Prediction of Systemic Lupus Erythematosus and Sjogren’s Syndrome Overlap"

_jcm, 2023, doi:10.3390/jcm12020535_

Round 1

Reviewer 1 Report

When you use new technique, there should be a reason why it was used. You proposed a new technique, machine learning for discrimination of SLE and SS. But the results were nothing new. You mentioned; “Clinical features of SS versus SLE-SS such as dry mouth, dry eye, and a unique auto-183 antibody profile associated with pSS were identified.” “In our study, we constructed a predictive model 201 with risk factors such as proteinuria, hematuria, hypoalbuminemia, and positive anti-202 dsDNA antibodies, suggesting that when SS patients present with these symptoms, then 203 they are more likely to have combined SLE.” Your conclusions are already well-known. Most of rheumatologist were suspicious of combined disease when patients showed specific clinical manifestations. They don’t need machine learning technique to solve these issue. I cannot find any clinical meaning of this study.

Reviewer 2 Report

Thank you for giving me the time to review your manuscript. This manuscript is interesting and scientifically meaningful for considering clinical decision models for predicting systemic lupus erythematosus and Sjogren’s syndrome overlap. Regarding the contents, the following revision should be considered.

The title should include the study design.

In the introduction, there is no paragraph writing. Each paragraph is too short. The author should focus on theory building, the problems, and research question paragraphs. The first paragraph should focus on the general information regarding mental health in primary care in international contexts. Moreover, the second and third paragraphs should introduce the research question as the theoretical and conceptual framework in international contexts and research questions.

The introduction should include the international contexts and research questions of this study.

The sample section of the method contains no descriptions regarding sample calculation.

The first paragraph of the discussion part should focus on the summary of the results.

This study should describe the limitation of sampling bias and the results' applicability to other settings, and the future investigation in the limitation part.

In the conclusion or discussion, the study’s strengths should be focused on international readers.

The conclusion section should be added based on the instruction of the authors.

Round 2

Reviewer 1 Report

Thank you for your revision

Reviewer 2 Report

The manuscript has been considerably improved. I think that this paper is suited for inclusion in our journal.